# A Comparison and Safety Evaluation of Micellar versus Standard Vitamin D_3_ Oral Supplementation in a Randomized, Double-Blind Human Pilot Study

**DOI:** 10.3390/nu16111573

**Published:** 2024-05-22

**Authors:** Julia Solnier, Chuck Chang, Yiming Zhang, Yun Chai Kuo, Min Du, Yoon Seok Roh, Janet See, Jennifer Brix, Roland J. Gahler, Tim Green, Simon Wood

**Affiliations:** 1ISURA, Clinical Research, Burnaby, BC V3N4S9, Canada; cchang@isura.ca (C.C.); yzhang@isura.ca (Y.Z.); rkuo@isura.ca (Y.C.K.); mdu@isura.ca (M.D.); kroh@isura.ca (Y.S.R.); 2Factors Group of Nutritional Companies Ltd., Burnaby, BC V3N4S9, Canada; jsee@factorsgroup.com (J.S.); jbrix@factorsgroup.com (J.B.);; 3Brix Wellness, Ltd., Victoria, BC V8Z 3E9, Canada; 4College of Nursing and Health Sciences, Flinders University, Sturt Road, Adelaide, SA 5042, Australia; tgreen@flinders.edu.au; 5School of Public Health, Faculty of Health Sciences, Curtin University, Perth, WA 6845, Australia; simonwood@shaw.ca; 6InovoBiologic Inc., Calgary, AB Y2N4Y7, Canada; 7Food, Nutrition and Health Program, University of British Columbia, Vancouver, BC V6T1Z4, Canada

**Keywords:** bioavailability, cholecalciferol, delivery systems, micelles, pharmacokinetics, supplementation, vitamin D

## Abstract

The aim of this pilot study was to evaluate and compare bioavailability and safety of two Vitamin D_3_ formulations (softgels) in healthy adults, at single daily doses of 1000 and 2500 IU, over a 60-day period. A total of 69 participants were initially screened for eligibility in a double-blind randomized study with a four-arm parallel design; 35 participants were randomized to treatment groups: (1) standard Vitamin D_3_ 1000 IU (STD1000), (2) micellar Vitamin D_3_ 1000 IU (LMD1000), (3) standard Vitamin D_3_ 2500 IU (STD2500), and (4) micellar Vitamin D_3_ 2500 IU (LMD2500). Serum Vitamin D concentrations were determined through calcifediol [25(OH)D] at baseline (=before treatment), at day 5, 10, and 15 (=during treatment), at day 30 (=end of treatment), and at day 45 and 60 (=during follow-up/post treatment). Safety markers and minerals were evaluated at baseline and at day 30 and day 60. The pharmacokinetic parameters with respect to iAUC were found to be significantly different between LMD1000 vs. STD1000: iAUC(5–60): 992 ± 260 vs. 177 ± 140 nmol day/L; *p* < 0.05, suggesting up to 6 times higher Vitamin D_3_ absorption of LMD when measured incrementally. During follow-up, participants in the LMD1000 treatment group showed approx. 7 times higher Vitamin D_3_ concentrations than the STD1000 group (iAUC(30–60): 680 ± 190 vs. 104 ± 91 nmol day/L; *p* < 0.05). However, no significant differences were found between the pharmacokinetics of the higher dosing groups STD2500 and LMD2500. No significant changes in serum 1,25(OH)_2_D concentrations or other biochemical safety markers were detected at day 60; no excess risks of hypercalcemia (i.e., total serum calcium > 2.63 mmol/L) or other adverse events were identified. LMD, a micellar delivery vehicle for microencapsulating Vitamin D_3_ (LipoMicel^®^), proved to be safe and only showed superior bioavailability when compared to standard Vitamin D at the lower dose of 1000 IU. This study has clinical trial registration: NCT05209425.

## 1. Introduction

Vitamin D is a hormone, best known for its role in calcium and phosphate metabolism related to bone health [1,2]. Beyond its traditional role in bone metabolism, Vitamin D has diverse physiological functions and is thought to be involved in the suppression of immune-mediated diseases, infections, certain cancers, and cardiovascular disease [3].

The two primary physiological forms of Vitamin D are cholecalciferol (Vitamin D_3_) and ergocalciferol (Vitamin D_2_) [4]. The latter is less bioavailable than Vitamin D_3_ and is only found naturally in fungi and a few other foods. Vitamin D_3_ can be obtained from a small number of fish (i.e., salmon) or fortified foods. Additionally, Vitamin D_3_ can also be endogenously synthesized from 7-dehydrocholesterol, a derivative of cholesterol; its production in the skin is stimulated upon exposure to Ultraviolet B radiation from sunlight. Both endogenously synthesized and exogenous Vitamin D must be converted to 25-hydroxycholecalciferol (25(OH)D_3_), a reaction catalyzed by the hepatic enzyme 1 alpha hydroxylase [5]. The circulating concentration of 25(OH)D represents the best biomarker of Vitamin D status [6]. Several cytochrome P-450 enzymes (CYPs), including CYP2R1, CYP27A1, and CYP2D25, are responsible for the conversion of Vitamin D to 25(OH)D [5,7].

Due to the small amounts obtained from the limited food sources, most people rely on endogenous synthesis for Vitamin D. Anything that affect the synthesis of Vitamin D in the skin, such as skin color and sun exposure, will affect Vitamin D status, the latter factor being influenced by season, geographic latitude, sunscreen use, and clothing [8,9,10]. For many individuals, especially those who avoid the sun or live at high latitudes in the winter months, it is difficult to maintain adequate 25(OH)D concentrations without the use of Vitamin D supplements [11,12]. Currently, there is no global consensus on optimal Vitamin D supplementation levels for overall health; thus, dosage recommendations vary among countries, ranging from 400 to 2000 IU Vitamin D per day (10–50 μg) [13,14]. Several large-scale randomized clinical trials (RCTs) have confirmed that higher daily dosages of Vitamin D (e.g., 4000 IU) are safe for adults and do not appear to cause significant problems such as arterial calcification or reduced kidney function in long-term studies (~2–5 years) with up to 30,000 participants [15,16,17,18,19,20].

Vitamin D_3_ is the main form of the vitamin used in supplements [21,22]. Different formulations of Vitamin D_3_ are available, such as those taken as oral drops, softgel capsules, or intramuscular injections [23,24]. Human studies investigating the influence of different supplemental forms on Vitamin D bioavailability are still sparse [23]. Limited clinical data are available on the pharmacokinetics of novel Vitamin D delivery vehicles [25,26,27,28,29] as well as on the impact of different dosing and treatment schedules on Vitamin D_3_ absorption in humans [30,31,32,33]. In these studies, the oral bioavailability of Vitamin D_3_ is generally evaluated through the major circulating metabolite, 25-hydroxyvitamin D (25(OH)D), which has a circulating half-life of 3 weeks [34] and thus serves as a stable biomarker of systemic Vitamin D status [35,36].

While altering the composition and daily dosage of a nutritional supplement may only minimally influence the absorption of Vitamin D_3_, changes in the formulation (i.e., delivery system) can significantly impact the bioavailability and metabolism of Vitamin D in the body. As a highly hydrophobic compound, characterized by a logP > 7, Vitamin D cannot freely penetrate inner body compartments; thus, it depends on carriers such as lipoproteins, specific receptors, and complex cellular redistribution mechanisms for effective transport [37]. Lipid-based carriers, like LipoMicel^®^ Vitamin D (LMD), can incorporate Vitamin D_3_ into their hydrophobic centers along with fatty acids and natural surfactants (e.g., lecithin), which aid in its emulsification and solubilization within the surfactant-stabilized aqueous solution. Compared to standard Vitamin D (STD) formulated in a liquid (oil-based) softgel, the formation of micellar Vitamin D particles in LMD may further facilitate its absorption in the jejunum and ileum, with presumably higher uptake through the lymphatic system [24]. Previous in vitro experiments have shown that LMD, microencapsulating and emulsifying Vitamin D_3_ to form micelles (d < 50 µm), achieved significantly higher Caco-2-cell permeability than standard Vitamin D_3_ [38].

In view of these findings, the goal of this (follow-up) pilot study was to compare micellar Vitamin D_3_ (i.e., LMD) with standard Vitamin D_3_ (i.e., STD), both formulated in a liquid (oil-based) softgel capsule, regarding their pharmacokinetic and safety profile in healthy adults following daily oral administration at dosages of 1000 IU and 2500 IU. Thereafter, the pharmacokinetics with respect to AUC and Cmax of serum circulatory 25-hydroxyvitamin D (25(OH)D), as well as 1,25-dihydroxyvitamin D (1,25(OH)_2_D) levels along with safety markers including mineral levels, were monitored over 60 days.

The hypotheses of this work were that LMD would raise serum circulatory 25(OH)D levels more efficiently than STD, and the daily supplementation with LMD would prove to be safe and effective for short-term use in healthy adults.

## 2. Materials and Methods

### 2.1. Study Design and Recruitment

This is a four-arm parallel double-blind randomized clinical trial. The study was conducted mainly during the Canadian winter months (Nov to April). Participants were recruited across British Columbia, Canada, through advertising, such as by study flyers.

To be eligible, participants had to be healthy and between 21 and 65 years. The following blood parameters were checked and had to be within normal ranges to participate in the study: alkaline phosphatase (ALP), alanine aminotransferase (ALT), aspartate aminotransferase (AST), total bilirubin, C reactive protein (CRP), creatinine (create), and gamma-glutamyl transferase (GGT). Participants had to refrain from any additional dietary supplements containing Vitamin D and allow for at least a 14-day washout prior to enrollment, as well as to refrain from any artificial UVB sources, including the use of tanning beds. Participants had to complete an online questionnaire on their medical history, weight, height, lifestyle (smoking, exercising, etc.), and dietary habits relating to food rich in Vitamin D, including other dietary supplementations.

Exclusion criteria were as follows: Vitamin D levels > 90 nmol/L (25(OH)D), the use of Vitamin D from 14 days prior to enrollment until the end of the study; the use of calcium, magnesium, fish oil, or omega 3 fatty acid supplements; any history of acute and/or chronic illness (such as gastrointestinal, liver, and kidney disorders, osteoporosis, etc.); and pregnancy or lactation. All participants had to provide their written informed consent before participating in this study.

Ethical approval was granted by the Institutional Review Board (IRB) of the Canadian SHIELD Ethics Review Board (OHRP registration IORG0003491; FDA registration IRB00004157; approval letter ID 2021-11-001, date of approval: 22 November 2021). The study has been registered on ClinicalTrials.gov with Identifier NCT05209425 and conducted per the ethical standards as outlined in the Helsinki Declaration of 1975. Figure 1 shows the flowchart of participants; the CONSORT reporting guidelines were used as a checklist for the present randomized trial.

### 2.2. Randomization and Blinding

Simple randomization was performed with Microsoft Excel’s (Version 2016) randomization function. The numbers in the randomized list were then subgrouped into 4 equal ranges, with each range representing one of the four treatments: regular/standard Vitamin D_3_ 1000 IU, LipoMicel^®^ Vitamin D_3_ 1000 IU, regular/standard Vitamin D_3_ 2500 IU, and LipoMicel^®^ Vitamin D_3_ 2500 IU. Next, the randomized list of treatments was numbered sequentially with a 3-digit code.

Capsules for each treatment were packaged in identical opaque bottles labeled according to the 3-digit code from the randomized list. Each participant who enrolled in the study and met the inclusion criteria was sequentially assigned one of the 3-digit codes from the randomization list and given the matching bottle of capsules. The randomization list was kept in a secure file by the study coordinator, and blinding was maintained throughout the study. Participants, assessors, and researchers were unaware of the treatment assignments until after the data were analyzed.

### 2.3. Study Protocol

The primary objective of this study was to compare the PKs of regular/standard Vitamin D_3_ (STD) with those of a new (microencapsulated) LipoMicel^®^ Vitamin D_3_ formulation (LMD) administered at single doses of 1000 IU and 2500 IU/day, respectively, in healthy volunteers. Serum Vitamin D_3_ concentrations were primarily determined through 25(OH)D and additionally through 1,25(OH)_2_D measurements upon enrollment, i.e., baseline (=before start of treatment), and were measured again during treatment at day 5, 10, and 15, at end of treatment at day 30, and during follow-up (= post treatment) at day 45 and 60.

The secondary objective of this study was to evaluate safety blood markers and mineral levels, which included serum levels of ALP, ALT, AST, BiliT, CRP, creatinine, and GGT. Mineral levels of calcium, magnesium, phosphate, potassium, and sodium were measured at baseline, end of treatment at day 30, and during follow-up (=post treatment) at day 60.

The entire study included a period of 60 days, with a 30-day treatment/supplementation period and a subsequent 30-day follow-up period (=free/post treatment).

In all four treatment groups, venous blood samples were collected for pharmacokinetic analysis pre-dose (=baseline), at day 5, 10, 15, and 30 during the treatment period, and at day 45 and 60 of the follow-up period. As for the safety analysis, venous blood samples were collected at pre-dose (=baseline), at day 30 at the end of the treatment, and at day 60 of the follow-up period.

Blood samples of participants were collected at each time point by a registered phlebotomist at different LifeLabs locations across British Columbia (BC), Canada (LifeLabs Medical Laboratory Services, Vancouver, BC, Canada). Samples were analyzed for 25(OH)D and 1,25(OH)_2_D using an ISO 15189 [39] accredited chemiluminescent immunoassay method.

### 2.4. Study Treatments

Two oral preparations of Vitamin D_3_ were procured from Natural Factors (Burnaby, BC, Canada). Each product in this study provided a total single dose of 1000 IU or 2500 IU Vitamin D_3_, formulated in soft gelatin capsules containing an oily solution of Vitamin D_3_; differences are found in the co-ingredients, e.g., the natural surfactants used to form micelles:-LMD: LipoMicel^®^ formulation was composed of Vitamin D_3_ (cholecalciferol) encapsulated with medium-chain triglycerides. Its micellular membrane contains food-grade excipients (patent pending). One capsule contained either 1000 IU or 2500 IU Vitamin D_3_ (cholecalciferol), as well as gelatin, glycerin, purified water (softgel), MCT, cocoa, xylitol, MSM (methylsulfonylmethane), and organic flaxseed oil.-STD: Standard/reference formulation was composed of Vitamin D_3_ (cholecalciferol). One capsule contained either 1000 IU or 2500 IU Vitamin D_3_ (cholecalciferol), as well as gelatin, glycerin, purified water (softgel), and organic flaxseed oil.

Treatments were administered orally each day at a single dose of either 1000 IU or 2500 IU Vitamin D_3_ with a glass of water (approx. 250 mL) and breakfast (optional) for a period of 30 days. Quality control testing was performed on all study/treatment samples; stability testing was conducted in accordance with ICH standards, and the products passed.

### 2.5. Statistical Analysis

Statistical analyses were performed with GraphPad Prism 10.2 (Boston, MA, USA). AUC’s were calculated using the trapezoid method. iAUC (incremental Area Under Curve) values were also calculated using the trapezoid method, but only the portion above baseline (initial value) was used. Considerations for missing data were made by using the imputed-mean and last-observation-carried-forward methods.

Comparison of the individual pharmacokinetic parameters of 25(OH)D and 1,25(OH)2D between the STD and LMD groups was made with multiple Mann–Whitney tests followed by Holm–Šídák multiple comparison correction. Differences were deemed significant at *p* ≤ 0.05.

For baseline characteristics, gender was evaluated with Fisher’s exact test for any significance in differences observed between the STD and LMD treatments. Evaluation of baseline characteristics other than gender was performed by first assessing the data for normality using the Shapiro–Wilk test. All parameters other than height, weight, BMI, and 1,25(OH)2D passed the test for normality at alpha = 0.05. Mann–Whitney tests followed by Holm–Šídák correction were used to determine the significance in differences between height, weight, BMI, and 1,25(OH)2D of the LMD and STD treatments. The remaining parameters were evaluated with multiple *t*-tests followed by Holm–Šídák correction.

Safety data for biochemical profiles were assessed for normality using the Kolmogorov–Smirnov test with an alpha value of 0.05. Non-normal data were logarithmically transformed and combined with the remaining parameters to be evaluated with repeated measures 2-way ANOVA with Dunnett correction.

## 3. Results

### 3.1. Baseline Characteristics

69 healthy volunteers were initially screened for eligibility. Of these, 35 subjects were randomized to treatment; ~74% were female and 26% were male, with a mean age of 41.1 years. Two participants were lost during follow-up; 33 participants completed the study as per protocol and were included in both the PK and safety analyses (see flow of the study, Figure 1).

The participants were all in good physical health, with safety markers within the normal ranges. Baseline demographic and biochemical characteristics are reported in Table 1. No significant differences between groups were observed at baseline for any of the reported parameters.

### 3.2. Pharmacokinetic Parameters

Significant differences were only found between STD1000 and LMD1000 with regard to the iAUC values (Table 2; Figure 1, Figure 2, Figure 3 and Figure 4. For example, when measured incrementally with baseline concentrations subtracted, the LMD1000 treatment group showed up to ∼6 times higher Vitamin D_3_ concentrations than the STD1000 group over the entire study period (iAUC(5–60), *p* < 0.05, Table 2). Similarly, during follow-up participants in the LMD1000 treatment group had approx. 7 times higher Vitamin D_3_ concentrations than the STD 1000 group (iAUC(30–60); *p* < 0.05, Table 2). However, no significant differences were found between the higher dosing groups, i.e., STD2500 and LMD2500 treatment, as illustrated in Figure 4a. Although LMD2500 showed a higher Cmax value, it did not reach statistical significance (Table 2).

As highlighted in Figure 2, the LMD1000 treatment group showed comparable average serum concentrations of 25(OH)D to the higher dosing STD2500 group, when measured incrementally. Additionally, changes in serum 1,25(OH)_2_D concentrations were monitored over 60 days; no significant differences were detected among the treatment groups when compared at the same dosages (Table 3; Figure 3 and Figure 4b).

There is typically no direct correlation in serum concentrations of 1,25(OH)_2_D and 25(OH)D in healthy individuals [6].

### 3.3. Changes in Safety Blood Markers

Significant reductions in ALT, AST, and GGT were observed in participants of the LMD2500 and STD1000 groups at day 30; however, these changes remained within normal ranges. No significant changes were found in any of the tested biochemical markers at the end of the study period (day 60; Table 4). Similarly, no significant changes in serum mineral levels for calcium (Ca), magnesium (Mg), phosphate (Phos), potassium (K), or sodium (Na) were detected in any treatment group over the study period (60 days).

### 3.4. Adverse Events

No adverse events (AEs) occurred during the study. None of the study participants reported any AEs. Of note, there were no recorded cases of hypercalcemia (i.e., total serum calcium > 2.63 mmol/L).

## 4. Discussion

Both formulations used in this study, STD and LMD, have been previously evaluated in terms of their in vitro characteristics, such as solubility, stability, and intestinal permeability [38]. The hypotheses of this study were based on the previous findings—assuming LMD, a new delivery system which microencapsulates Vitamin D_3_ (LipoMicel^®^ Vitamin D), would show superiority over STD in terms of its pharmacokinetic profile in healthy adults and that oral supplementation with either Vitamin D_3_ formulation, STD or LMD, would prove to be safe in participants at a daily administration of two different dosages (1000 IU and 2500 IU) over a study period of 60 days.

In contrast to other studies, this work monitored changes in both 25(OH)D as well as 1,25(OH)_2_D serum concentrations up to 30 days post treatment in order to obtain a more comprehensive understanding of the Vitamin D absorption and metabolism in study participants after stopping supplementation.

Results demonstrated that only micellar Vitamin D_3_ at a dose of 1000 IU (LMD1000) achieved significantly higher serum 25(OH)D concentrations in participants compared to the standard Vitamin D_3_ (STD1000) treatment. No significant differences were detected between the higher dosing groups, LMD2500 and STD2500. Notably, participants of each group started with different baseline mean 25(OH)D values, ranging from 39.3 ± 9.6 to 71.6 ± 29.4 nmol/L (Table 1). Yet, when measured incrementally, with baseline concentrations subtracted, LMD1000 achieved higher absorption of Vitamin D_3_ (approx. 6 times higher iAUC(5–60), *p* < 0.05) compared to the same dose of STD1000. Even after the end of treatment, the LMD1000 group showed approx. 7 times higher 25(OH)D concentrations than the STD1000 group (iAUC(30–60); Table 2). Interestingly, LMD1000 illustrated comparable serum 25(OH)D concentrations to STD2500 when measured incrementally, even though it contained a dosage of Vitamin D that is 2.5 times lower (Figure 2).

Results of this research are in accordance with another study on liposomal Vitamin D_3_ that reported four times greater AUC values of 25(OH)D when compared to a standard oil-based Vitamin D_3_ formulation [26]. However, a much higher daily dose of 10,000 IU of Vitamin D_3_ was administered in that study compared to the doses used in the current study. Furthermore, while other studies showed earlier maximum/peak serum 25(OH)D concentrations following oral Vitamin D_3_ administration [26,27,40,41], both STD and LMD demonstrated peak 25(OH)D concentrations at the end (~30 days) or after the supplementation period (~40 days). For instance, the highest, although not significant, 25(OH)D concentrations were demonstrated by LMD2500 (Cmax 98.4 ± 16 nmol/L) around day 32. The prolonged time to reach Cmax may be a more common pharmacokinetic characteristic of higher doses and novel formulations of Vitamin D since Mentaverri et al. also reported a similar delay in participants reaching a Cmax of 28.5 ± 5.0 ng/mL (71.1 ± 12.5 nmol/L)—around 14 days—when administered in a higher single dose of 100,000 IU [27]. Because this study included non-deficient individuals with higher baseline Vitamin D concentrations, a slower response, i.e., a more gradual rise in 25 (OH)D serum levels, would typically be expected [41], as opposed to the faster/steeper increase in 25 (OH)D serum levels (i.e., earlier Tmax) typically observed in deficient participants with low baseline values [27]. Similar to the higher dosing Vitamin D groups, LMD1000 showed significantly higher 25 (OH)D serum levels during follow-up (i.e., post treatment) when compared to the same dose of STD (iAUC(30–60); Table 2). This suggests the storage of Vitamin D_3_ in body fat when serum Vitamin D_3_ concentrations are elevated; it further indicates that the entire quantity of ingested Vitamin D_3_ is not immediately metabolized as the hepatic 25-hydroxylases become saturated. Instead, it reaches equilibrium in the body fat, from where it is gradually released into the circulation and subsequently metabolized into 25(OH)D_3_, even after the end of supplementation.

In anticipation of these findings, changes in safety parameters, including serum 1,25(OH)_2_D, were analyzed throughout the study period. Serum 25-hydroxyvitamin D concentrations above 375 nmol/L (>150 ng/mL) would indicate Vitamin D overdosing and a high risk of hypercalcemia [42]. In addition, it is especially desirable to measure both metabolites—25-hydroxyvitamin D (25(OH)D) and 1,25-dihydroxyvitamin D (1,25(OH)_2_D)—in the diagnosis of hypercalcemia [43]. In this study, none of the participants exhibited serum 25-hydroxyvitamin D concentrations deemed concerning (>100 nmol/L), and no instances of hypercalcemia were detected. In fact, both high-absorption groups (LMD1000 and LMD2500) demonstrated minimal changes in calcium concentrations throughout the study. Therefore, this study suggests that the use of higher doses, including high-bioavailability formulations of Vitamin D_3_ such as LMD2500, is not associated with elevated 25(OH)D serum levels posing a potential risk for hypercalcemia (>2.63 mmol/L). While certain liver markers, specifically ALT and AST, in the LMD2500 group exhibited a temporary significant (though still within normal ranges) reduction compared to baseline, they reverted to non-significant levels by day 60, marking the conclusion of the study period. This aligns with other study findings demonstrating that the prolonged daily intake of high doses of Vitamin D (e.g., 10,000 IU) did not result in any adverse effects in healthy adults [19,27,41]. Serum levels of 1,25(OH)_2_D, the active form of Vitamin D, can be indicative of the overall metabolism of Vitamin D (e.g., hormonal feedback regulation), as well as of the safety and responsiveness towards the treatments (e.g., genetic polymorphisms in cytochrome P450 enzymes) [37]. The production of 1,25(OH)_2_D is tightly regulated based on calcium and phosphorus levels, and excess amounts could lead to hypercalcemia. In fact, when there is sufficient Vitamin D, the body may suppress the production of 1,25(OH)_2_D to prevent excess calcium absorption. Thus, other important safety markers to consider along with the 1,25(OH)_2_D) concentrations include calcium, and phosphorus (which is interconnected with calcium homeostasis), as well as magnesium levels and kidney markers. Results of this study showed no significant changes in 1,25(OH)_2_D or the other safety markers. This conclusion is also supported by other published studies [19,20]. Furthermore, magnesium concentrations were similar between the treatment groups and did not change significantly over the study period, otherwise the absorption efficiency of Vitamin D could have been impacted [44,45].

Only a few human studies have investigated the absorption characteristics of different novel Vitamin D_3_ formulations. Most of these studies have administered a single megadose instead of the daily, comparably lower doses administered in the current work. For instance, Marwaha et al. reported higher bioavailability of a Vitamin D_3_ nanoemulsion (oral solution) at 60,000 IU with approx. 1.4 times higher mean Cmax values compared to standard softgel capsules (127.10 ± 26.24 ng/mL vs. 91.82 ± 29.27 ng/mL), followed for 5 days post administration [46]. Another similar study found that the bioavailability of Vitamin D_3_ softgel capsules administered at 100,000 IU was superior to that of a standard oral solution (Cmax: 28.5 ± 5.0 ng/mL vs. 23.9 ± 4.3 ng/mL), followed for 112 days post administration [27]. Bano et al. compared three different Vitamin D_3_ formulations administered at the same dose but using different treatment schedules in deficient adults. The authors concluded that a new orodispersible Vitamin D_3_ formulation (sucrosomial^®^) achieved approximately twice the 25(OH)D levels compared to a chewable tablet administered weekly for 3 weeks at a dose of 200,000 IU (78.8 ± 7.1 vs. 36.1 ± 3.6 ng/mL) and outperformed softgel capsules administered every other week for 6 weeks (66.1 ± 5.1 vs. 38.2 ± 2.6 ng/mL) [47]. Radicioni et al. found only slightly higher bioavailability of a new orally disintegrating film formulation of Vitamin D_3_ when compared to standard oral Vitamin D_3_ solution following a single dose of 25,000 IU, with Cmax values of 6.68 ± 2.03 vs. 6.61 ± 2.62 ng/mL [48].

In comparison, the LMD2500 formulation used in this study achieved a higher Cmax of 98.4 ± 16 nmol/L over a shorter period of 60 days despite being administered at a lower dose of 2500 IU per day or 75,000 IU cumulatively over 30 days.

However, when compared to the equivalent dose of the standard (STD) treatment, this study demonstrates the non-inferiority of the LMD2500 product. Further investigation through larger scale studies and extended follow-up periods may be necessary to ascertain any significant differences.

It is especially advantageous to examine a lower dose of Vitamin D_3_ in the context of nutritional supplementation, which is typically provided to consumers as over-the-counter products without a prescription. From different regulatory bodies (for example, Health Canada [49]), nutritional supplementation receives tight regulation, and higher doses fall into the category of prescription medication, which can be more difficult to obtain by an average consumer for the purposes of routinely boosting their Vitamin D levels. This work revealed that lower doses of Vitamin D_3_ in the LMD formulations are both safe and pharmacokinetically efficacious at a dose of 1000 IU.

Micellar delivery vehicles consisting of spherical particles within the micrometer size range (approximately 1 to 50 µm in diameter, as exemplified by LMD [38]) may offer advantages for cellular uptake and may exhibit higher thermodynamic stability compared to nanoscale liposomes. Also, micelles may be more resistant to changes in temperature and pH and have a lower tendency to aggregate or coalesce compared to smaller nanoparticles like liposomes [50,51,52]. This leads to increased stability and shelf life, which are important factors when considering the industrial scale production of the resulting Vitamin D formulations for commercial purposes [53].

This work contributes valuable insights into the PKs and safety of oral Vitamin D supplementation, with a focus on innovative delivery systems for optimizing serum 25(OH)D levels. However, certain limitations of this study should be acknowledged. For instance, the small sample size of this pilot trial may affect the generalizability of the findings applicable to a larger adult population. Therefore, the results of this research would benefit from confirmation in a future, larger clinical study.

Also, this study was conducted on healthy individuals only; thus, the results may not be generalizable to a diseased population. In fact, study participants started with mean 25(OH)D baseline levels of 51.6 ± 23.2 nmol/L, which is within the normal range recommended for overall health. However, in the context of this study, more substantial differences in elevating 25(OH)D levels would probably have been observed in deficient patients (<30 nmol/L) of a larger sample size. Especially for future intervention trials, only participants with Vitamin D deficiency should be included in order to observe significant benefits of novel delivery systems, like LMD, on clinical outcomes. Furthermore, future follow-up studies may explore earlier time points concerning Vitamin D absorption (e.g., AUC(0–48)). Finally, the authors suggest that future large-scale RCTs should investigate the enhanced absorption of novel Vitamin D interventions across diverse subgroups. These may include variations in age, BMI, ethnicity, gender, underlying health conditions such as malabsorption, and genetic factors such as polymorphisms in Vitamin D-related genes like the Vitamin D receptor (VDR) and cytochrome P450 enzymes, including CYP2R1, CYP27B1, CYP24A1, and CYP27A1, among others.

## 5. Conclusions

This study concludes that LMD (LipoMicel^®^ Vitamin D), specifically at a dose of 1000 IU, is a promising intervention to effectively raise 25(OH)D serum levels in human participants, with superior bioavailability compared to the same dose of standard Vitamin D_3_ formulation (STD). Participants supplementing with LMD at a dose of 1000 IU had significantly higher incremental 25(OH)D serum concentrations (~7 times) than the STD group at 30 days post treatment (iAUC(30–60)). Non-inferiority of the LMD2500 product was shown when compared to the equivalent dose of the standard (STD) treatment.

The supplementation of Vitamin D_3_, up to doses of 2500 IU, appeared to be safe and well tolerated, irrespective of the delivery form. None of the participants exhibited serum 25(OH)D concentrations that raised concerns (>100 nmol/L), and, specifically, no significant changes in calcium and phosphate levels were found, including no occurrence of hypercalcemia (>2.63 mmol/L).

## 6. Patents

Pending for LipoMicel^®^ Matrix—Eutectic Matrix for Nutraceutical Compositions lists inventors as: R.J.G., S.W., Y.C.K. and C.C. No inventor benefits from this, and the ownership belongs to InovoBiologic Inc. (Calgary, AB, Canada).

## Figures and Tables

**Figure 1 nutrients-16-01573-f001:**
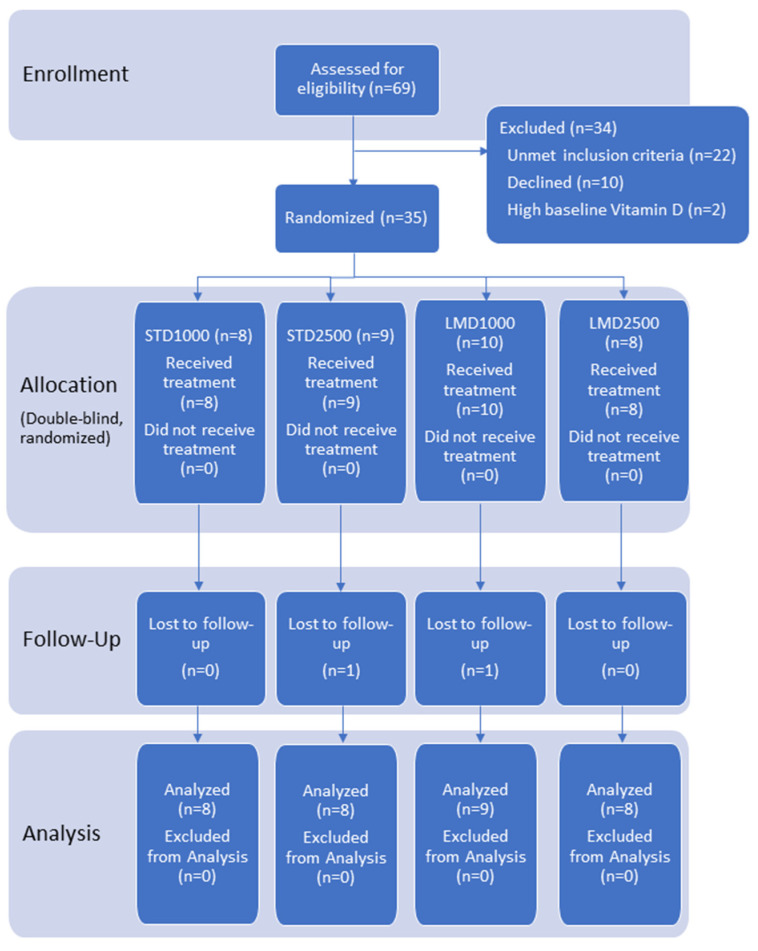
Participant flow chart.

**Figure 2 nutrients-16-01573-f002:**
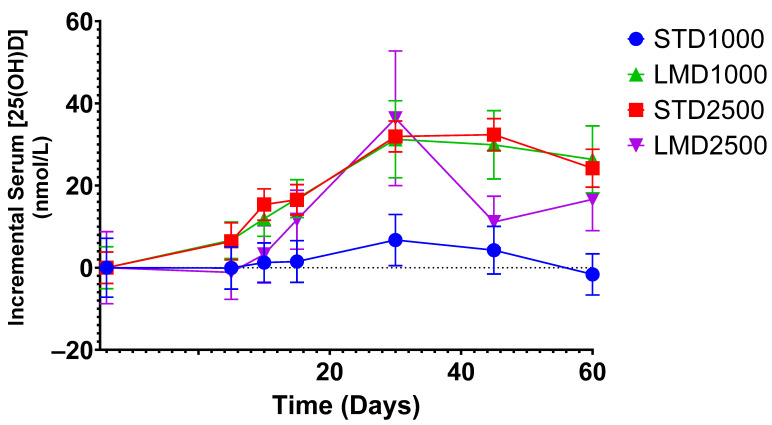
Incremental average serum concentrations of 25(OH)D in healthy participants after baseline subtraction at various time points. Mean ± SEM; significance was calculated using the Mann–Whitney test with Holm–Šídák multiple comparison correction.

**Figure 3 nutrients-16-01573-f003:**
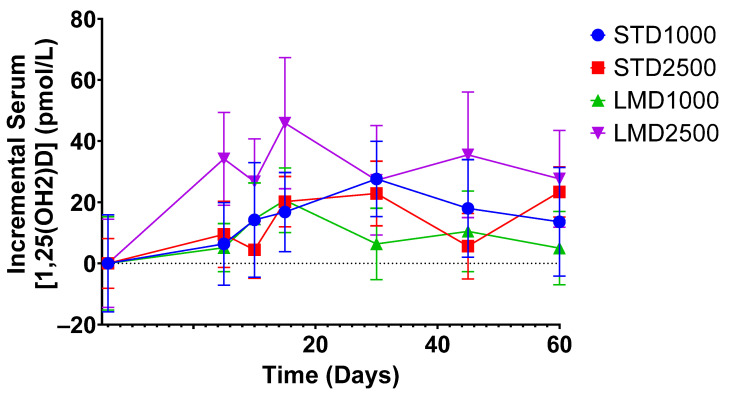
Incremental average serum concentrations of 1,25(OH)_2_D in healthy participants at various time points. Mean ± SEM; significance was calculated using the Mann–Whitney test with Holm–Šídák multiple comparison correction.

**Figure 4 nutrients-16-01573-f004:**
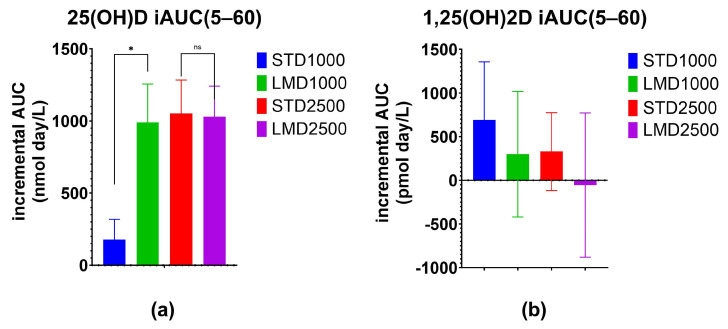
Increase in serum (**a**) 25(OH)D and (**b**) 1,25(OH)2D concentrations over the 60-day study period. Incremental Area Under Curve (iAUC) values obtained from subtraction of baseline values for each participant at various time points were evaluated. Mean ± SEM; * denotes *p* ≤ 0.05 using the Mann–Whitney test with Holm–Šídák multiple comparison correction; ns denotes “not significant”.

**Table 1 nutrients-16-01573-t001:** Baseline characteristics.

Participants Characteristics	All Subjects *n* = 35	STD1000 *n* = 8	LMD1000 *n* = 10	STD2500 *n* = 9	LMD2500 *n* = 8
Age, years	41.1 ± 1.8	37.6 ± 11.8	40.3 ± 7.6	45.9 ± 3.9	40.4 ± 3.7
Gender					
Male, *n* (%)	9 (25.7)	1 (12.5)	2 (20)	3 (33.3)	3 (37.5)
Female, *n* (%)	26 (74.3)	7 (87.5)	8 (80)	6 (66.7)	5 (62.5)
Weight, kg	67.3 ± 2.6	66 ± 4.2	64 ± 4.6	62.7 ± 2.9	77.8 ± 7.3
BMI, kg/m^2^	24.1 ± 1.0	25.2 ± 1.4	21.9 ± 2.4	23.6 ± 1.4	26.1 ± 2.0
25(OH)D, nmol/L	49 ± 3.2	54.4 ± 6.7	43.3 ± 4.4	39.3 ± 3.2	61.9 ± 8.2
1,25(OH)_2_D, pmol/L	133.1 ± 6.4	138 ± 15	138 ± 13.3	126.6 ± 9.2	128.9 ± 12.5
BiliT, µmol/L	7.17 ± 0.59	6.25 ± 0.70	10.3 ± 1.2	5.33 ± 0.96	6.6 ± 0.8
ALP, U/L	75.5 ± 3.6	64.9 ± 5.5	79.7 ± 6.0	84.4 ± 9.4	70.1 ± 3.7
GGT, U/L	17.6 ± 1.8	14 ± 1.7	21.2 ± 4.7	17.7 ± 3.2	16.8 ± 2.5
ALT, U/L	22.2 ± 3.1	17.1 ± 3.5	27.6 ± 9.2	20.8 ± 3.1	22.4 ± 4.2
AST, U/L	19.1 ± 0.7	16.4 ± 1.3	19.6 ± 1.4	20.8 ± 1.3	20 ± 1.5
CRP, mg/L	1.46 ± 0.20	1.68 ± 0.46	2.1 ± 0.49	1.51 ± 0.18	1.7 ± 0.5
Na, mmol/L	139.9 ± 0.3	140.2 ± 0.8	139.4 ± 0.5	140.1 ± 0.6	139.8 ± 0.5
K, mmol/L	4.26 ± 0.05	4.10 ± 0.07	4.22 ± 0.14	4.37 ± 0.07	4.35 ± 0.07
Ca, mmol/L	2.38 ± 0.02	2.40 ± 0.04	2.36 ± 0.03	2.34 ± 0.02	2.41 ± 0.03
Phos, mmol/L	1.10 ± 0.03	1.14 ± 0.06	1.10 ± 0.05	1.06 ± 0.05	1.10 ± 0.05
Mg, mmol/L	0.84 ± 0.01	0.79 ± 0.01	0.85 ± 0.02	0.86 ± 0.01	0.83 ± 0.02

No statistically significant difference between STD and LMD groups; gender was evaluated with Fisher’s exact test; height, weight, BMI, and 1,25(OH)2D were evaluated with Mann–Whitney tests followed by Holm–Šídák correction; the remaining parameters were evaluated with multiple t-tests followed by Holm–Šídák correction; BMI = body mass index; data presented as mean ± standard error of mean or number (%).

**Table 2 nutrients-16-01573-t002:** Pharmacokinetic parameters for serum 25(OH)D concentrations over 60 days.

25(OH)D	1000 IU		2500 IU	
	STD1000 *n* = 8	LMD1000 *n* = 9	STD2500 *n* = 8	LMD2500 *n* = 8
iAUC(5–30) [nmol day/L]	73.3 ± 53	312 ± 78	337 ± 64	431 ± 84
iAUC(5–45) [nmol day/L]	157 ± 93 *	670 ± 180 *	724 ± 140	804 ± 180
iAUC(5–60) [nmol day/L]	177 ± 140 *	992 ± 260 *	1050 ± 230	1030 ± 210
iAUC(30–60) [nmol day/L]	104 ± 91 *	680 ± 190 *	715 ± 170	598 ± 130
iAUC(15–45) [nmol day/L]	146 ± 80 *	619 ± 170 *	654 ± 130	750 ± 170
AUC(0–60) [nmol day/L]	3440 ± 330	3930 ± 380	3790 ± 180	4680 ± 520
Cmax [nmol/L]	61.6 ± 6	78.3 ± 9.3	75.1 ± 3.8	98.4 ± 16
Tmax [days]	26.3 ± 4.5	40 ± 4.3	32.5 ± 4.7	31.9 ± 4.4

Means ± SEM reported. * denotes *p* ≤ 0.05 using the Mann–Whitney test with Holm–Šídák multiple comparison correction.

**Table 3 nutrients-16-01573-t003:** Pharmacokinetic parameters for serum 1,25(OH)_2_D concentrations over 60 days.

1,25(OH)_2_D	1000 IU		2500 IU	
	STD1000 *n* = 8	LMD1000 *n* = 9	STD2500 *n* = 8	LMD2500 *n* = 8
iAUC(5–30) [pmol day/L]	300 ± 330	210 ± 320	180 ± 150	27 ± 300
iAUC(5–45) [pmol day/L]	550 ± 500	260 ± 530	250 ± 310	−16 ± 570
iAUC(5–60) [pmol day/L]	690 ± 670	300 ± 720	330 ± 450	−55 ± 830
iAUC(30–60) [pmol day/L]	390 ± 370	130 ± 380	150 ± 310	−82 ± 560
iAUC(15–45) [pmol day/L]	480 ± 420	170 ± 440	250 ± 260	−7.5 ± 490
AUC(0–60) [pmol day/L]	9340 ± 730	9090 ± 460	8070 ± 360	9600 ± 930
Cmax [nmol/L]	190 ± 13	180 ± 7.2	160 ± 7.6	200 ± 15
Tmax [days]	23 ± 4.6	24 ± 6.3	29 ± 6.9	23 ± 6.7

Means ± SEM reported. Significance was calculated using the Mann–Whitney test with Holm–Šídák multiple comparison correction. No significance was found between STD1000 and LMD1000 nor between STD2500 and LMD2500.

**Table 4 nutrients-16-01573-t004:** Changes (%) in safety biochemical profile from baseline.

	STD1000		LMD1000		STD2500		LMD2500	
	30-Day	60-Day	30-Day	60-Day	30-Day	60-Day	30-Day	60-Day
Na	0.6 ± 0.3	0.1 ± 0.5	0.7 ± 0.4	0.5 ± 0.4	4.5 ± 4	−0.4 ± 0.6	0.6 ± 0.5	0.9 ± 0.7
K	1 ± 1.4	0.6 ± 1	−4 ± 1.4	−1.7 ± 4	0.4 ± 2.1	−2.5 ± 2.8	−0.7 ± 2.7	−1.8 ± 3.1
Creat	−2.4 ± 3.3	−2.9 ± 5	3.7 ± 3.1	8 ± 8.7	2.8 ± 5.1	0.7 ± 5	−0.6 ± 3.1	0.4 ± 3.6
GFR	6 ± 5	−1.4 ± 4.1	−0.7 ± 2.5	2.1 ± 5.2	−0.5 ± 5.1	1.1 ± 5.1	2 ± 4.4	3.4 ± 4.2
Ca	−0.4 ± 1.1	−1.4 ± 1.1	2.4 ± 1.2	0.4 ± 1.3	−0.5 ± 0.4	0 ± 1.1	0.2 ± 1.2	−2.3 ± 1.4
Phos	−4.1 ± 5.7	2 ± 2.2	0.9 ± 5.5	−2.7 ± 5.9	20 ± 8.7	6.2 ± 5.5	7.8 ± 9.8	13 ± 7.8
BiliT	−12 ± 12	8.3 ± 23	21 ± 14	−16 ± 9.7	60 ± 27	26 ± 10	12 ± 9.9	28 ± 24
ALP	−0.6 ± 2.1	0.9 ± 2.2	0.6 ± 1.3	10 ± 9.2	3.1 ± 5.8	7.4 ± 4.9	1.9 ± 1.8	1.3 ± 3.4
GGT	−8 ± 2.3 *	−2.6 ± 3.7	6.4 ± 4.4	17 ± 21	11 ± 18	15 ± 19	1 ± 5.3	5.7 ± 5.9
ALT	14 ± 16	15 ± 17	−12 ± 11	−0.6 ± 30	−5.9 ± 11	−8.6 ± 14	−12 ± 2.9 *	−13 ± 7.8
AST	11 ± 10	16 ± 15	9.1 ± 12	11 ± 15	6.6 ± 6.5	1.7 ± 9.1	−11 ± 3.9 *	−8.2 ± 6.6
Mg	2.8 ± 2.1	1.6 ± 3.2	0.6 ± 1.3	2.2 ± 1.6	−2.7 ± 2.2	−3.6 ± 2.3	−4.9 ± 1.6	3.7 ± 2.9
CRP	−22 ± 12	−22 ± 8.2	28 ± 50	40 ± 47	1.9 ± 15	−2.5 ± 18	11 ± 28	77 ± 51

Means ± SEM reported. * denotes *p* ≤ 0.05 using repeated measures 2-way ANOVA with Dunnett correction for multiple comparisons against baseline.

## Data Availability

Data and/or statistical analyses are available upon request due to trade secrets, privacy, legal or ethical reasons.

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
