# Peer review of "A Comparison and Safety Evaluation of Micellar versus Standard Vitamin D3 Oral Supplementation in a Randomized, Double-Blind Human Pilot Study"

_nutrients, 2024, doi:10.3390/nu16111573_

Round 1

Reviewer 1 Report (New Reviewer)

Comments and Suggestions for Authors

Interesting paper, in general well-written. Only a few points:

1.      Any power calculation? The current sample size seems to be quite small to me

2.      Also due to the small sample size, Table 1 shows the characteristics between comparison groups are not balanced (the difference may not reach statistical significance due to power), including important potential effect modifiers like BMI

3.      From which source the trial participants were recruited? Why omega 3 fatty acids supplements users were excluded? Would this kind of exclusion influence the generalizability of the study findings?

Author Response

We would like to thank the Reviewers for their valuable feedback. Please see our responses below. All changes are highlighted in the manuscript (e.g., see “track changes”). 

REVIEWER1:

Comments and Suggestions for Authors

Interesting paper, in general well-written. Only a few points:

  1. Any power calculation? The current sample size seems to be quite small to me.

You are correct, the sample size is small.  However, our study was designed as a pilot investigation and as such we felt after discussion with a statistician (PhD level), that we would design it as per the information mentioned in our paper.

  1. Also due to the small sample size, Table 1 shows the characteristics between comparison groups are not balanced (the difference may not reach statistical significance due to power), including important potential effect modifiers like BMI.

We were aware of this small sample size but as a pilot study we wanted to at first see how the product behaved per study design. We did, however, include various parameters that we thought were important to measure using blood testing.  We still contend that despite these limitations, valuable insights were gained regarding the feasibility and acceptability of the intervention, which should encourage us to do further research using larger subject numbers.

We did highlight several limitations of this pilot study – Lines 461-476. For example, we mentioned that “… future large-scale RCTs should investigate the enhanced absorption of novel Vitamin D interventions across diverse subgroups. These may include variations in age, BMI (=newly added; Line 476), ethnicity, gender, underlying health conditions such as malabsorption, and genetic factors such as polymorphisms in vitamin D-related genes like the vitamin D receptor (VDR), cytochrome P450 enzymes including CYP2R1, CYP27B1, CYP24A1, CYP27A1, among others.”

  1. From which source the trial participants were recruited?

Study participants were recruited across British Columbia, Canada, through advertising such as study flyers. Please see Line 114.

Why omega 3 fatty acids supplements users were excluded? Would this kind of exclusion influence the generalizability of the study findings?

The relationship between omega-3 fatty acid supplements and vitamin D absorption is complex and not fully understood. However, there is some evidence to suggest that omega-3 fatty acids may have some effect on vitamin D metabolism and absorption1,2,3. For example, omega-3 fatty acids may enhance the bioavailability of vitamin D by improving its absorption in the gut1.

1 Walia, N., Dasgupta, N., Ranjan, S., Chen, L., & Ramalingam, C. (2017). Fish oil based vitamin D nanoencapsulation by ultrasonication and bioaccessibility analysis in simulated gastro-intestinal tract. Ultrasonics Sonochemistry, 39, 623-635. https://doi.org/10.1016/j.ultsonch.2017.05.021

2 Al-Shaer AH, Abu-Samak MS, Hasoun LZ, Mohammad BA, Basheti IA. Assessing the effect of omega-3 fatty acid combined with vitamin D3 versus vitamin D3 alone on estradiol levels: a randomized, placebo-controlled trial in females with vitamin D deficiency. Clin Pharmacol. 2019;11:25-37. https://doi.org/10.2147/CPAA.S182927

3 Laing, B. B., Cavadino, A., Ellett, S., & Ferguson, L. R. (2020). Effects of an Omega-3 and Vitamin D Supplement on Fatty Acids and Vitamin D Serum Levels in Double-Blinded, Randomized, Controlled Trials in Healthy and Crohn’s Disease Populations. Nutrients, 12(4). https://doi.org/10.3390/nu12041139

Reviewer 2 Report (New Reviewer)

Comments and Suggestions for Authors

the size of the groups is small to allow firm conclusions

the delta 25OHD after  regular 1000 IU D3 is rather small in comparison with other studies dealing with subjects with rather   poor vit  D status at baseline

it is most unfortunate that serum vitamin D itself within 48 h after first dose was not  measured as this would really low whether the observed differences are due to different in absorption or transport or metabolism

Author Response

We would like to thank the Reviewers for their valuable feedback. Please see our responses below. All changes are highlighted in the manuscript (e.g., see “track changes”).   

REVIEWER2:

Comments and Suggestions for Authors

  1. The size of the groups is small to allow firm conclusions.

You are correct, the sample size is small.  However, our study was designed as a pilot investigation and as such we felt after discussion with a statistician (PhD level), that we would design it as per the information mentioned in our paper. Furthermore, we did acknowledge all limitations of the study in our discussion. For example, please see Lines 462-465:

However, certain limitations of this study should be acknowledged. For instance, the small sample size of this pilot trial may affect the generalizability of the findings applicable to a larger adult population. Therefore, the results of this research would benefit from confirmation in a future, larger clinical study.”

  1. The delta 25OHD after regular 1000 IU D3 is rather small in comparison with other studies dealing with subjects with rather poor Vit D status at baseline.

Compared to other studies using Vitamin D deficient subjects reporting a larger change in 25OHD after supplementation, in this pilot trial participants started with rather high Vit D baseline levels of 51.6 ± 23.2 nmol/L - as mentioned in the discussion, please see Lines 467-471. Therefore, we mention that “… in the context of this study, more substantial differences in elevating 25(OH)D levels would probably have been observed in deficient patients (<30 nmol/L) of a larger sample size.”

  1. It is most unfortunate that serum vitamin D itself within 48 h after first dose was not measured as this would really show whether the observed differences are due to differences in absorption or transport or metabolism.

In general, similar studies have suggested that significant increases in serum vitamin D levels can be observed within a few days to weeks after starting supplementation (all studies are referenced in the paper). We assumed that it is highly unlikely to see noticeable changes in serum vitamin D levels within just 48 hours after taking the first dose of a vitamin D supplement. Therefore, the rationale of this study design was based on previous research.

However, we recognize the potential for larger follow-up clinical studies to explore earlier time points concerning vitamin D absorption. We added this to our discussion, please see Line 474.

Round 2

Reviewer 1 Report (New Reviewer)

Comments and Suggestions for Authors

Thanks for the revision. I have no more comments.

This manuscript is a resubmission of an earlier submission. The following is a list of the peer review reports and author responses from that submission.

Round 1

Reviewer 1 Report

Comments and Suggestions for Authors

Solnier et al. aimed to evaluate the bioavailability and safety of micellar versus standard vitamin D3 oral supplementation in healthy adults, at single daily doses of 1000 and 2500 IU, over a 60-day period by conducting a double-blind randomized study with a four-arm parallel design. The results indicate that LMD (LipoMice Vitamin D) is a promising intervention to effectively raise 25(OH)D serum levels in human participants with up to 6-times higher bioavailability compared to standard Vit D3 formulation (STD). This suggests that smaller doses of LMD can achieve the same vitamin D blood concentrations as a higher dose of standard treatment. Supplementation of vitamin D3, up to doses of 2500 IU, appeared to be safe and well-tolerated, irrespective of the delivery form. This study is important as it demonstrates the superior bioavailability of the micellar delivery vehicle for microencapsulating Vit D3 (LipoMice) compared to standard Vit D supplement.

Major points: In Figure 2, STD2500 appears to have the same pharmacokinetics (PK) as LMD1000. LMD2500 seems to have less bioavailability compared to STD2500 and LMD1000 except on day 30. However, in Table 2, the area under the curve (AUC) (0-45 or 0-60) of LMD2500, but not LMD1000 or STD2500, was demonstrated to have significantly higher AUCs than STD1000. It feels to me that there might be a contradiction between the impression conveyed by Figure 2 and the statistical significance presented in Table 2. It's possible that I'm misunderstanding something fundamental, but could you please provide an explanation that clarifies this issue?

Minor points:

Figure 1 should be explained in the results section.

In Table 1, 20 is better expressed as 20.0.

In Table 2, *denotes p<0.05… is better written as footnotes under the table.

In Table 3, "No significance between…" is the same as above.

Author Response

We would like to thank the Reviewers for their valuable feedback. Please see our responses below. All changes are highlighted in the manuscript (e.g., see “track changes”).

Major points:

  • In Figure 2, STD2500 appears to have the same pharmacokinetics (PK) as LMD1000. LMD2500 seems to have less bioavailability compared to STD2500 and LMD1000 except on day 30. However, in Table 2, the area under the curve (AUC) (0-45 or 0-60) of LMD2500, but not LMD1000 or STD2500, was demonstrated to have significantly higher AUCs than STD1000. It feels to me that there might be a contradiction between the impression conveyed by Figure 2 and the statistical significance presented in Table 2. It's possible that I'm misunderstanding something fundamental, but could you please provide an explanation that clarifies this issue?

Thank you for pointing this out! To improve the clarity of our data presentation, we have re-arranged the tables to now compare the same dosages with each other; we also included an extra bar graph comparing LMD 1000 vs STD 1000 and LMD2500 vs STD 2500 (Figure 4). Figure 2 demonstrates the incremental serum concentrations (baseline subtracted) of 25(OH)D whose area under the curve would correspond to iAUC’s instead of AUC’s in Table 2. The Figure 2 caption has been amended to state Incremental average serum concentrations of 25(OH)D in healthy participants after baseline subtraction at various time points."

Originally, Table 2 compared all treatments to STD1000; this has been amended so that the same dosages of the two formulations are compared to each other. Therefore, the statistical analysis has been updated.

Minor points:

  • Figure 1 should be explained in the results section.

Please see in the revised results section (3.1. Baseline Characteristics): “In total, 69 healthy volunteers were initially screened for eligibility. Of these, 35 subjects were randomized to treatment; ~74% were female and 26% were male with a mean age of 41.1 years. 33 participants completed the study as per protocol and were included in both the PK- and safety analyses (see flow of the study; Figure 1).”

  • In Table 1, 20 is better expressed as 20.0.
  • In Table 2, *denotes p<0.05… is better written as footnotes under the table.

In Table 3, "No significance between…" is the same as above.

Please note that all tables have been revised due to updated statistics. Please see the revised tables in the manuscript.

Reviewer 2 Report

Comments and Suggestions for Authors

In this manuscript the authors compare two different formulations of oral Vitamin D (each one with two different concentrations) for a period of 30 days.

This is an interesting study although with exaggerated conclusions.

When comparing things, the authors need to be fair on their comparisons. Each formulation should be compared with the other formulation but with the same concentration. The 1000 with the 1000 and the 2500 with the 2500. To compare 1000 with the 2500 does not make sense unless we want to say that 2500 is better.

And statistical analysis exists to be used and interpretation of results should be based on statistical differences. To state that one is different than the other but without statistical significance shouldn’t be done. If the difference was not statistically significant, there was no difference.

The LMD2500 group had the highest Vid D levels at the beginning. Could that have influenced the analysis?

The study showed a non-inferiority of the LMDproduct. And that is reasonable. Maybe with a larger study and longer follow-up a difference can be found.

Can the authors explain the lower value of the 25 Vit D levels at day 45 in the LMD2500 with subsequent rise at day 60?

A statistician should review the manuscript.

Author Response

We would like to thank the Reviewers for their valuable feedback. Please see our responses below. All changes are highlighted in the manuscript (e.g., see “track changes”).

In this manuscript the authors compare two different formulations of oral Vitamin D (each one with two different concentrations) for a period of 30 days. This is an interesting study although with exaggerated conclusions.

Major points:

  • When comparing things, the authors need to be fair on their comparisons. Each formulation should be compared with the other formulation but with the same concentration. The 1000 with the 1000 and the 2500 with the 2500. To compare 1000 with the 2500 does not make sense unless we want to say that 2500 is better.

We appreciate the recommendation. - To improve the clarity of our data presentation, we have now arranged the tables to compare the same dosages with each other (i.e., LMD 1000 vs STD 1000 and LMD 2500 vs 2500 STD); we also included an extra bar graph comparing LMD 1000 vs STD 1000 and LMD2500 vs STD 2500 (Figure 4). We have amended all tables to show comparisons between the same dosages.

  • Statistical analysis exists to be used and interpretation of results should be based on statistical differences. To state that one is different than the other but without statistical significance shouldn’t be done. If the difference was not statistically significant, there was no difference.

The statistical analysis has been updated accordingly and the interpretation of the results as well as the conclusions have been based on statistically significant outcomes.

  • The LMD2500 group had the highest Vid D levels at the beginning. Could that have influenced the analysis?

We have now focused on baseline subtracted incremental results (e.g., iAUC) to reduce the influence of higher initial values. This has been further addressed in the discussion, Line 346-350: (Although not significant) “participants of each group started with different baseline mean 25(OH)D values, ranging from 39.3 ± 9.6 — 71.6 ± 29.4 nmol/L (Table 1). Yet, when measured incrementally, with baseline concentrations subtracted, LMD1000 achieved the highest absorption of Vitamin D3 (approx. 6 times higher iAUC5-60, p<0.05) compared to the same dose of STD1000.”

The study showed a non-inferiority of the LMD product. And that is reasonable. Maybe with a larger study and longer follow-up a difference can be found.

We have adjusted the discussion based on the updated stat. analysis. For example, please see Line 431-435 “When compared to the equivalent dose of the standard (STD) treatment, this study demonstrates the non-inferiority of the LMD2500 product; further investigation through larger-scale studies and extended follow-up periods may be necessary to ascertain any significant differences.”

  • Can the authors explain the lower value of the 25 Vit D levels at day 45 in the LMD2500 with a subsequent rise at day 60? There is no significance in the difference between the 25(OH)D concentrations of LMD2500 at Day 45 and Day 60 since the error bars overlap with each other. A statistician should review the manuscript.

We have performed a Repeat Measures One-way ANOVA on the data for LMD2500 with Tukey’s multiple comparisons test. The significance of the difference in values between Day 45 and Day 60 has a p-value of 0.5542. This is in line with the overlapping measurement error ranges for the two timepoints displayed in Figure 2. However, compared to the other three treatments, LMD2500 did appear to show reduced serum 25(OH)D levels in a quicker, though non-significant, manner that may be worthy of future investigation.

Please see the updated manuscript - statistical analysis has been reviewed and reevaluated accordingly.
